# Learning to Walk from Three Minutes of Real-World Data with Semi-structured Dynamics Models

**Jacob Levy**[*]
University of Texas at Austin
jake.levy@utexas.edu

**Tyler Westenbroek**[*]
University of Washington
westenbroekt@gmail.com

**David Fridovich-Keil**
University of Texas at Austin
dfk@utexas.edu

**Abstract:** Traditionally, model-based reinforcement learning (MBRL) methods exploit neural networks as flexible function approximators to represent *a priori* unknown environment dynamics. However, training data are typically scarce in practice, and these black-box models often fail to generalize. Modeling architectures that leverage known physics can substantially reduce the complexity of system-identification, but break down in the face of complex phenomena such as contact. We introduce a novel framework for learning semi-structured dynamics models for contact-rich systems which seamlessly integrates structured first principles modeling techniques with black-box auto-regressive models. Specifically, we develop an ensemble of probabilistic models to estimate external forces, conditioned on historical observations and actions, and integrate these predictions using known Lagrangian dynamics. With this semi-structured approach, we can make accurate long-horizon predictions with substantially less data than prior methods. We leverage this capability and propose Semi-Structured Reinforcement Learning (SSRL) a simple model-based learning framework which pushes the sample complexity boundary for real-world learning. We validate our approach on a real-world Unitree Go1 quadruped robot, learning dynamic gaits – from scratch – on both hard and soft surfaces with just a few minutes of real-world data. Video and code are available at: https://sites.google.com/utexas.edu/ssrl

**Keywords:** Model-Based Reinforcement Learning, Physics-Based Models

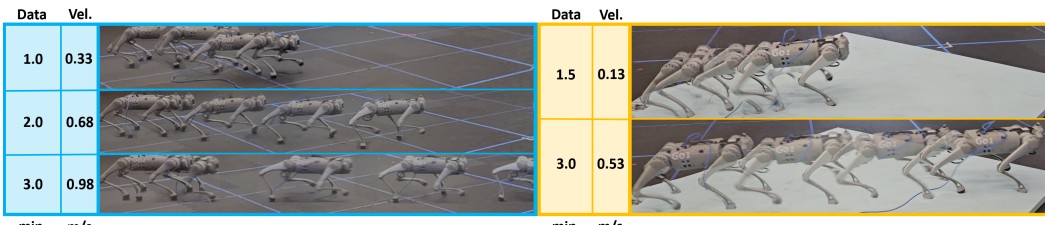

Figure 1: Unitree Go1 quadruped learning to walk from scratch using SSRL on hard ground (left) and memory foam (right).

## 1 Introduction

Effective robotic agents must leverage complex interactions between the robot and its environment, which are difficult to model using first principles. Model-based reinforcement learning (MBRL) is a powerful paradigm for controller synthesis [1], wherein the robot learns a generative dynamics model for the environment. The model can then be used to hallucinate synthetic rollouts [2], providing a source of data augmentation for policy optimization algorithms [3, 4, 5]. When the model is accurate, it can generate long rollouts which extrapolate beyond the training data, accelerating policy learning substantially. However, in practice, the black-box neural network models favored in the

---

[*]These authors contributed equally.

8th Conference on Robot Learning (CoRL 2024), Munich, Germany.

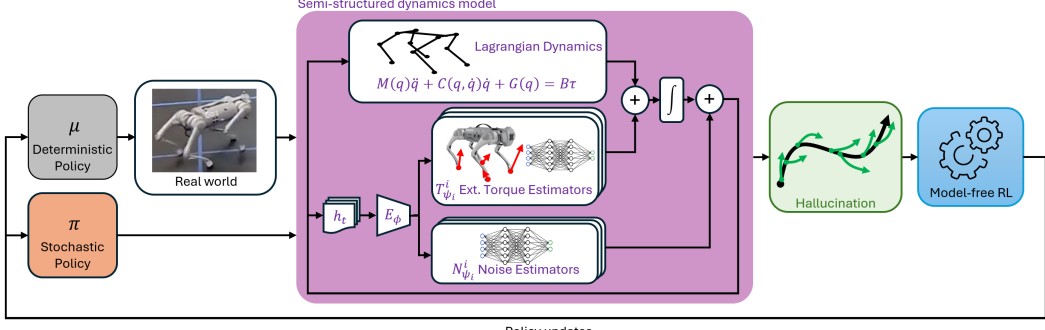

Figure 2: The SSRL framework. A deterministic policy is used to collect data from the real world while a stochastic policy is utilized in conjunction with the learned dynamics model to "hallucinate" short synthetic rollouts which branch from this data. The model incorporates Lagrangian dynamics and encodes previous state predictions, which are fed to external torque and noise estimators to predict future states. The synthetic data is used with a model-free RL algorithm to update the policies.

MBRL literature struggle to generalize beyond the training data [6, 7, 8], and thus do not outperform modern model-free alternatives [9, 10]. Currently, both paradigms are too inefficient and unreliable to make learning new behaviors in the real world practical for many applications.

An appealing alternative is to leverage known physics to design structured model classes. This general approach has been used for efficient system identification and controller synthesis across many bodies of work, ranging from classic adaptive control techniques [11, 12, 13, 14] to more recent physics-informed neural architectures [15, 16, 17]. However, these previous approaches do not scale to the complexities of real-world learning for contact rich-systems such as walking robots (Fig. 1). Indeed, modeling contact remains an open problem. Moreover, prior structured approaches to learning contact dynamics make strong assumptions about what can be perceived with available on-board sensors. For example, at inference time [18] requires access to a signed-distance representation of potential contact surfaces. More generally, prior approaches generally assume access to privileged state observations which make it possible to predict exactly when and where contact is made [19, 15]. However, reliably estimating these quantities in the real-world using noisy on-board sensors is another open area of research [20, 21, 22, 23, 24].

We ask: *can we build a light-weight modeling framework which leverages known structure and is implementable with onboard sensors?* We answer this question in the affirmative by introducing Semi-structured Reinforcement Learning (SSRL), a simple MBRL pipeline for contact-rich control in the real-world (Fig. 2). For concreteness, we focus primarily on the quadruped depicted in Fig. 1 and aim to learn an effective locomotion controller **entirely from scratch in the real world**. We consider the case where the robot's observations only include proprioceptive measurements via joint encoders, IMU measurements, and a global velocity estimator.

We inspire our approach by looking to the Lagrangian equation of motion for the robot:

$$M(q)\ddot{q} + C(q, \dot{q}) + G(q) = B\tau + J^T F^e + \tau^d, \tag{1}$$

where $q$ and $\dot{q}$ are the generalized coordinates and velocities of the robot, motor torques $\tau$ are distributed to the joints by the matrix $B$, $F^e$ represents contact forces generated by the environment with $J$ the resulting Jacobian, $\tau^d$ represents parasitic damping torques, and $M$, $C$, and $G$ are the mass matrix, Coriolis and centrifugal forces, and gravity vector. Here, $B, M, C,$ and $G$ are determined by the geometry and inertial properties of the robot, which are known *a priori*. Thus, given access to observations, we assume that the only unknown terms are $J^T F^e$ and $\tau^d$.

We leverage the known Lagrangian structure of (1) by estimating these unknown terms with an estimator $\hat{\tau}^e(h_t) \approx J^T F^e + \tau^d$ which is conditioned on a history $h_t$ of the available measurements, enabling the model to infer information about hidden states of the environment such as the geometry of the ground underfoot [25, 26]. We instantiate $\hat{\tau}^e$ as an ensemble of probabilistic models to help

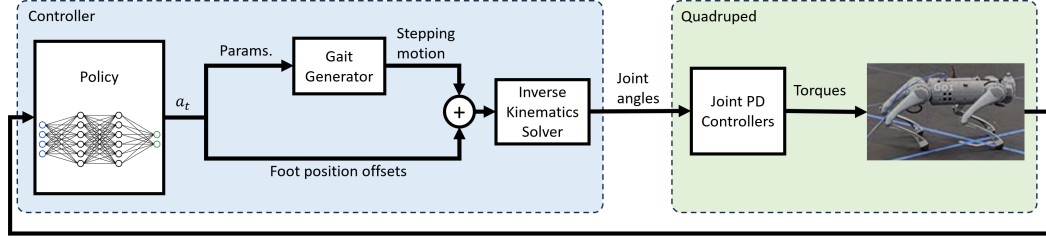

Figure 3: Control architecture. The policy takes in a history of observations and outputs parameters to a gait generator and offsets to the gait. The resulting foot positions are sent to an inverse kinematics solver which computes desired joint angles for joint level PD controllers.

capture uncertainty in the predictions, as is commonplace in the MBRL literature [6, 27]. We then introduce a novel inference procedure for making forward predictions, wherein the predictions made by these semi-structured models are fed back into $\hat{\tau}^e$ in an auto-regressive fashion. The SSRL framework leverages these predictions as a source of data augmentation to accelerate real-world policy optimization, using a simple Dyna-style [3, 6] MBRL algorithm. Finally, we demonstrate that this light-weight approach scales to the real-world, learning to walk on different terrains within a minute and learning substantially more dynamic gates within a few minutes. This result outperforms recent state-of-the-art real-world learning results for quadrupeds [28] by both $a$) requiring an order-of-magnitude less real-world data and $b$) reaching substantially higher walking speeds.

## 2  Preliminaries and Problem Formulation

**Notation:** While the underlying dynamics of the robot evolve in continuous time according to the differential equation (1), the policy acts in discrete time and we will use subscripts to denote discrete time steps. The state of the robot $s_t \in \mathcal{S}$ captures the available proprioceptive states $q_t$ and their velocities $\dot{q}_t$, available from joint encoders and IMU measurements. This does not include global coordinates of the robot. We capture the state of the environment (or *extrinsics*) $e_t \in \mathcal{E}$, which includes non-proprioceptive information (such as the distance of the ground below the robot) and the state of the ground underfoot (such as terrain deformation). The robot actions $a_t \in \mathcal{A}$ are the outputs of neural network described below. The overall dynamics for the system are defined by:

$$\text{Robot Transitions: } s_{t+1} \sim p_s(\cdot|s_t, a_t, e_t) \qquad \text{Environment Transitions: } e_{t+1} \sim p_e(\cdot|s_t, a_t, e_t). \quad (2)$$

To control this joint system, we will denote the policy the agent optimizes via $a_t \sim \pi_\theta(\cdot|s_t, h_t)$, where $h_t = (s_{t-1}, \ldots, s_{t-h})$ bundles the histories of state measurements and $\theta$ are parameters.

**Control Architecture:** We adopt the control architecture depicted in Fig. 3, similar to prior works on RL-based locomotion [25, 29, 30]. We optimize a neural network policy $\pi_\theta(\cdot|s_t, h_t)$ conditioned on previous observations which outputs (i) changes to parameters of a nominal gait generator that outputs desired foot positions and (ii) offsets to these foot positions. The resulting targets are sent to an inverse kinematics solver which computes desired joint positions. The joint targets are output at $100\,\text{Hz}$ to low-level PD controllers for conversion to torques. Additional details are provided in Appendix A.

**Reinforcement Learning Problem:** We formally frame the learning of a locomotion controller in the real world in terms of a partially observable Markov decision process (POMDP) [31], defined by the tuple $(\mathcal{X}, \mathcal{A}, p, r, \Omega, O, \gamma)$. Here, $\mathcal{X} = \mathcal{S} \times \mathcal{E}$ is the overall state space for the system, and $p(\cdot|s_t, a_t, e_t) = (p_s(\cdot|s_t, a_t, e_t), p_e(\cdot|s_t, a_t, e_t))$ captures the joint robot-environment dynamics. The space of observations $\Omega$ consists of the states that can be measured, and the observation distribution $O(\cdot|s_t, a_t, e_t)$ provides (noisy) estimates of the states from onboard sensors. The reward function $r(s_t, a_t, s_{t+1})$ depends only upon the robot states and actions, and is therefore directly measurable in the real world. We define the reward function to maximize the robot's forward velocity, maintain upright orientation, minimize angular rates, conserve energy, and avoid excessive torques.

Finally, we define a termination flag $d_t \in \{0, 1\}$ where $d_t = 1$ when body roll or pitch exceed limits. Exact definitions are found in Appendix A. Given an episode length $T \in \mathbb{N}$, discount factor $\gamma \in (0, 1)$, and a distribution $x_0$ over $\mathcal{X}$ of initial conditions for the system, the goal is to maximize the expected discounted total reward: $\max_\theta \mathbb{E}[\sum_{t=0}^{T} \gamma^t (1 - d_t) \cdot r(s_t, a_t, s_{t+1})]$.

# 3 Semi-structured Reinforcement Learning

A high-level overview of our method is presented in Fig. 2, outlined as follows: i) we learn an ensemble of deterministic external torque estimators $\hat{\tau}^e$ which are conditioned on $h_t$; ii) we then integrate the torque predictions through the Lagrangian ODE (1), add a learned noise term to generate probabilistic 1-step predictions, and feed this prediction back into $h_t$ to produce auto-regressive predictions over multiple steps iii) we fit this semi-structured representation of the dynamics using multi-step prediction losses; and finally iv) predictions from these models are used as a source of data augmentation for MBRL.

## 3.1 External Torque Estimators

We construct our approximations to the discrete-time probabilistic robot transition dynamics $s_{t+1} \sim p_s(\cdot|s_t, a_t, e_t)$ by building on top of the deterministic Lagrangian dynamics (1) which we rewrite as:

$$M(q)\ddot{q} + C(q, \dot{q}) + G(q) = B\tau + \underbrace{J(s, e)^T F^e(s, e, \tau) + \tau^d(s, e)}_{\tau^e(s, e, \tau)}, \tag{3}$$

where we now explicitly denote the dependence of the contact Jacobian $J(s, e)$, contact forces $F^e(s, e, \tau)$, and dissipative terms $\tau^d(s, e)$ on the state of the robot, environment, and low-level torques supplied by the motor. As discussed in Section 1, identifying precise locations where contact occurs on the robot can be extremely difficult from on-board measurements. Thus, instead of inferring $J$ and $F^e$ separately, we directly estimate $\tau^e = J^T F^e + \tau^d$. We condition these estimates a latent encoding $z_t = E_\phi(h_t)$ of $h_t$, which enables the network to infer information about the extrinsics $e_t$ needed for predicting future states [26, 25]. Altogether, we use an ensemble of deterministic models $\tau^e \approx \bar{\tau}_t^{e,i} = T_{\psi_i}(s_t, a_t, z_t)$ with tunable parameters $\psi_i$.

## 3.2 Generating Forwards Predictions

We first describe how we generate a probabilistic 1-step prediction for the next state associated to each of the torques estimates $\tau^e \approx \bar{\tau}_t^{e,i} = T_{\psi_i}(s_t, a_t, z_t)$. First, we let $\tau_t = G(s_t, a_t)$ denote a zero-order hold estimate for the low-level motor torques applied to the robot; here, $G$ is a known map which captures how the current state and action are processed by our control architecture (Fig. 3) to produce low-level joint torques. The $i$-th deterministic next-state prediction is then given by $\bar{s}_{t+1}^i = I(s_t, \tau_t, \bar{\tau}_t^{e,i})$, which captures how a chosen numerical integrator for the Lagrangian dynamics (1) propagates $s_t$, $\tau_t$, and $\bar{\tau}_t^{e,i}$. We then add zero-mean Gaussian noise to $\bar{s}_{t+1}^i$ whose variance $\Sigma_t^i = N_{\psi_i}^i(s_t, a_t, z_t)$ is a diagonal Gaussian which is output by the same network predicting the $i$-th torque estimate. Altogether, this constructs a probabilistic estimate for the next state:

$$\text{Uncertainty-Aware State Predictions:} \quad \hat{s}_{t+1}^i \sim \hat{p}_{\psi_i}^i(\cdot|s_t, a_t, h_t) := \mathcal{N}(\bar{s}_{t+1}^i, \Sigma_t^i) \tag{4}$$

The ensemble of probabilistic models $\{\hat{p}_{\psi_i}^i\}$ is used to generate $k$-step predictions using Algorithm 1, which adapts the method from [6] to our auto-regressive setting where future state predictions are conditioned on the history $h_t$. Given a state $s_t$ and state history $h_t$, to generate a synthetic rollout we: (i) sample an action from the policy, (ii) randomly choose a model $\hat{p}_{\psi_i}^i$ from the ensemble, (iii) generate a next-state prediction $\hat{s}_{t+1}^i$ using (4), and iv) incorporate this prediction into $h_{t+1}$ and repeat the previous steps until a $k$-step prediction has been generated.

## 3.3 Approximate Maximum Likelihood Estimation

To train $\{\hat{p}_\psi^i\}_i$, we maximize the joint-likelihood of state predictions along these synthetic rollouts under the real-world data. Due to the Markov assumption (Section 2), the joint distribution of

**Algorithm 1** Auto-Regressive State Predictions

---
1: **Inputs** hallucination buffer $\mathcal{D}_{\texttt{model}}$, models $\{\hat{p}_{\psi_i}^i\}$, policy $\pi_\theta$, start state $s_0$, start history $h_0$
2: **for** $t = 0 \ldots k - 1$ **do**
3:      Sample action $a_t \sim \pi_\theta(\cdot \mid s_t, h_t)$
4:      Randomly choose model $i \sim \mathcal{U}[1, \ldots, P]$ and predict next state $\hat{s}_{t+1}^i$ with (4)
5:      Update state history $h_{t+1}$ with $\hat{s}_{t+1}^i$ and $s_{t+1} \leftarrow \hat{s}_{t+1}^i$
6:      Compute reward $r_t$ and termination $d_t$ and add transition $(s_t, a_t, r_t, s_{t+1}, d_t)$ to $\mathcal{D}_{\texttt{model}}$
7: **Return** $\mathcal{D}_{\texttt{model}}$

---

the next $H$ states, starting at state $s_t$ is $p_s(s_{t+1:t+1+H} \mid s_t, a_{t:t+H}, e_{t:t+H}) = \prod_{j=0}^H p_s(s_{t+1+j} \mid s_{t+j}, a_{t+j}, e_{t+j})$. Taking the negative log-likllihood yields our training objective for the $i$-th model:

$$\mathcal{L}(\psi_i) = \frac{1}{HN_{\mathrm{e}}} \sum_{t=h}^{N_{\mathrm{e}}-H} \sum_{j=0}^H \left[\bar{s}_{t+1+j}^i - s_{t+1+j}\right]^\mathsf{T} \left(\Sigma_{t+j}^i\right)^{-1} \left[\bar{s}_{t+1+j}^i - s_{t+1+j}\right] + \log \det \Sigma_{t+j}^i. \quad (5)$$

Here, $N_{\mathrm{e}}$ is the size of a buffer of real-world transitions $\mathcal{D}_{\texttt{env}}$, mean state predictions are propagated deterministically according to $\bar{s}_{t+1}^i = S^i(\bar{s}_t^i, a_t, h_t)$, and each prediction $\bar{s}_t^i$ is used to update the state history $h_t$. For each state $s_t$ in the buffer, (5) generates a synthetic rollout $H$ steps long and computes a loss related to how much the propagated states differ from the experienced states.

**Remark 1.** *By optimizing the multi-step loss* (5) *end-to-end, we $i$) force the latent $z_t$ to encode information about the environment which is necessary for making state predictions over long horizons and $ii$) average out noisy state estimates and sudden changes in the external torques that occur when the robot makes contact with the ground (Fig. 4). This leads to reliable long-horizon predictions which are substantially more accurate than those generated by black-box models (Fig. 5).*

## 3.4 Policy Optimization

Finally, we introduce the Semi-Structured Reinforcement Learning (SSRL) in Algorithm 2. This policy optimization strategy is a Dyna-style algorithm [3, 6] which leverages model-predictions as a source of data augmentation. Specifically, this approach branches hallucinated rollouts off real-world trajectories to reduce the effects of model bias, and increases the length of these rollouts throughout training as the model becomes more accurate. We perform many rounds of data generation and policy optimization after collecting real-world samples, and as depicted in Fig. 5 our semi-structured model are able to make substantially more accurate predictions over long horizons when data is scarce. Together, these features enable SSRL to push the sample complexity limits for MBRL. We also found the following necessary for making real-world learning practical:

**Deterministic Real-World Rollouts and Random Hallucinations.** Steps in the real environment are taken using a deterministic policy $\mu_\theta$ which simply outputs the mean action from the stochastic policy $\pi_\theta$. Black-box MBRL approaches typically need to inject random 'dithering' noise to ensure their is enough exploration to obtain accurate models. However, we found that this led to unacceptably erratic behavior in the real-world when paired with our policy class. Recent model-free approaches for learning locomotion in the real-world [32] severely restrict the action space to make stochastic exploration tractable, and thus severely limit the performance of resulting policies. However, our semi-structured models have the right amount of inductive bias to accurately identify the dynamics when the real-world data set $\mathcal{D}_{\texttt{env}}$ is generated with deterministic exploration and data is scarce. We do, however, use a stochastic policy $\pi_\theta$ to generate rich, diverse synthetic data sets $\mathcal{D}_{\texttt{model}}$ for policy optimization. Specifically, policy parameters $\theta$ are trained with soft-actor critic (SAC) [33], using a mixture of transitions from real-world and hallucination buffers $\mathcal{D}_{\texttt{env}}$ and $\mathcal{D}_{\texttt{model}}$.

**Algorithm 2** SSRL : Policy Optimization with Semi-structured Dynamics Models

1: **Initialize** models $\hat{p}_{\psi_i}$, policy $\pi_\theta$, critics $Q_{\phi_i}$
2: **for** $N_{\text{epochs}}$ epochs **do**
3:     Take $N_E$ steps in the environment deterministically with $\mu_\theta$ and add transitions to $\mathcal{D}_{\text{env}}$
4:     Train models $\hat{p}_{\psi_i}$ on $\mathcal{D}_{\text{env}}$ using loss (5)
5:     Increase hallucination rollout length $k$ according to predetermined schedule
6:     **for** $K$ hallucination updates **do**
7:         **for** $M$ model rollouts **do**
8:             Sample state $s_0$ uniformly from $\mathcal{D}_{\text{env}}$ and hallucinate with Algorithm 1
9:         Perform $G$ updates of policy $\pi_\theta$ using mixture of $\mathcal{D}_{\text{env}}$ and $\mathcal{D}_{\text{model}}$ at ratio $r_\mathcal{D}$

## 4 Experimental Results

### 4.1 Real-world Results

We demonstrate our approach through two real-world experiments where a Unitree Go1 quadruped is trained from scratch to achieve maximum speed on both hard ground and memory foam.

**Experimental setup.** Training is performed from scratch in the real world over $N_{\text{epochs}} = 18$ epochs with $N_E = 1000$ environment steps per epoch, totaling $18,000$ steps or $3.0\,\text{min}$ of real-world interaction. We use an observation history length of $h = 5$, a multi-step loss horizon of $H = 4$, and hallucinate synthetic rollouts for up to $k = 20$ steps; see Appendix D for hyperparameters. To enhance plasticity, the model, actor, and critic are reset at $10,000$ steps [34]. Joint states are measured via encoders, IMU data provides body orientation and angular velocity, and linear velocity is acquired with a Vicon motion capture system. Neural networks are trained in JAX [35], and low-level joint angle commands are sent via Unitree's ROS interface [36]. We compute the mass matrix, Coriolis terms, and gravity vector in (3) using the differentiable simulator Brax [37], enabling gradient-based training of the semi-structured dynamics model with (5).

**Results.** Fig. 1 shows a time-lapse of rollouts as training progresses. After just $3.0\,\text{min}$ of real-world data, the quadruped achieves an average velocity of $0.98\,\text{m s}^{-1}$ on hard ground and $0.53\,\text{m s}^{-1}$ on memory foam (Fig. 4, right). Figure 4 (left) plots the reward per episode until the first termination ($d_t = 1$). Initially, the quadruped often falls, resulting in low rewards, but after $1.5\,\text{min}$, the learned policy becomes robust, and rewards increase. Despite the challenge of walking on memory foam, where the robot's feet sink deeply, forward velocity improves consistently in both terrains, demonstrating our approach's versatility to differing contact dynamics. To examine model accuracy, we compare the learned external force predictions $\bar{\tau}_t^{e,i}$ to real-world external force estimates $\tau^e$ over one second. We estimate the real-world external force with $\tau^e = M(q_t)\ddot{q}_t + C(q_t, \ddot{q}_t) + G(q_t) - B\tau_t$ where joint accelerations are estimated by finite differencing: $\ddot{q}_t = (\dot{q}_{t+1} - \dot{q}_t)/\Delta t$ and motor torques $\tau_t$ are estimated with the PD control law. Figure 4 (right) shows the learned external vertical force predicted on the base of the robot. Notably, the predictions appear as smoothed versions of the actual torque estimates. This, when combined with the accuracy of our predictions over long-horizons (Section 4.2) provides insight into why our approach enables such effective policy optimization [38]. Appendix B.1 provides plots for the force estimated on additional degrees of freedom for the robot.

### 4.2 Simulated Experiments

In addition to the results presented here, we provide extensive ablations on standard RL benchmarks in Appendix C. Here, we investigate the following hypotheses:

**Hypothesis 1.** *SSRL boosts performance and sample efficiency for MBRL in contact-rich settings.*

**Hypothesis 2.** *Training with a multi-step loss outperforms a single-step loss.*

**Hypothesis 3.** *Predictions from semi-structured dynamics models demonstrate greater accuracy and improved generalization beyond training data compared to black-box models.*

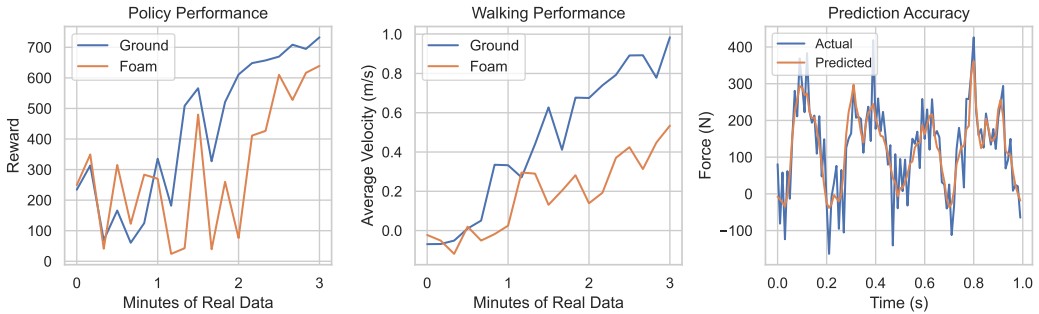

Figure 4: Real-world results. Left—SSRL efficiently performs policy optimization, even when data is scarce. Center—With our approach, the quadruped steadily learns to walk faster. Right—Predicted and real external vertical force acting on the robot base over one second of real-world data. Real forces are estimated by finite differences. The predictions add noticeable smoothing to the real-world data.

**Experimental setup.** The simulation setup mirrors the real-world setup (Section 4.1), except rollouts are simulated in Brax [37]; parameters are detailed in Appendix D. To examine Hypothesis 1, we perform policy optimization with SSRL (Algorithm 2) and compare its policy performance to a baseline approach that utilizes the same optimization process but substitutes the semi-structured dynamics model with a black-box model. We also benchmark against SAC [33], with an extended run shown in Appendix B.6. To test Hypothesis 2, we repeat the prior experiment, except we train the model with a single-step loss horizon ($H = 1$). To assess generalization (Hypothesis 3), we train our semi-structured models and the black-box models from scratch over 3 minutes of saved simulated data using 1- and 4-step losses. We then generate new data using a stochastic policy where we lower the friction and ground contact stiffness by $25\%$ in the simulator. We generate 20-step synthetic rollouts from 400 randomly-sampled starting states within the new dataset and average the prediction error $\|\hat{s}_t - s_t\|/\dim(s_t)$ over the 400 rollouts. All runs are repeated across 4 random seeds.

**Results.** Shown in Fig. 5 (left), our semi-structured dynamics models outperform black-box models in both sample efficiency and maximum reward, supporting Hypothesis 1. Even when data is scarce, our semi-structured model generates more accurate synthetic rollouts during exploration, resulting in significantly improved policy performance. We also observe in Fig. 5 (left) that, while black-box models yield similar policy performance for both single- and multi-step losses, training our semi-structured dynamics models with a multi-step loss results in improved performance over a single-step loss, confirming Hypothesis 2. In Fig. 5 (right), we see that our semi-structured models produce predictions 20 steps into the future that are significantly more accurate than black-box models. By leveraging physics-based knowledge, our models generate synthetic rollouts that generalize better to an unseen environment, confirming Hypothesis 3; additional experiments are found in Appendix B.

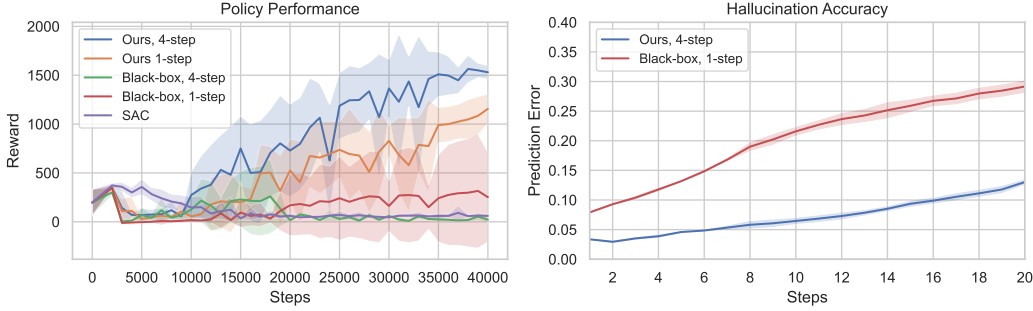

Figure 5: Left—SSRL achieves better policy performance compared to a baseline using black-box models. Right—Prediction error for 20-step synthetic rollouts in an unseen environment showcases our method's superior ability to generalize.

# 5 Related Work

**Model-based RL.** Our works builds on a wealth of prior works that use general function approximators and probabilistic modeling for black-box modeling [39, 6, 27]. Model-based reinforcement learning algorithms either learn a model that is used for online planning [40, 27, 41] or Dyna-style algorithms which hallucinate imagined rollouts for direct policy optimization [6, 3, 4]. Moreover, significant attention been devoted to extending these strategies to learn latent representations for environments from available observations [2, 42]. However, these approaches do not leverage known structure to accelerate learning. In contrast, there has been a substantial line of work which leverages known structure to efficiently learn accurate models [15, 16, 17, 19, 15]. However, these approaches generally require access to the full state of the system at inference time, which is impractical for real-world learning for contact-rich with on-board perception. Thus, our semi-structured approach brings the strengths of both paradigms to bear in a single framework. Finally, we also note related work [43, 44, 38] which investigates smoothing out contact dynamics to make policy optimization more tractable for difficult contact-rich problems. We found that our learned models naturally learned smooth representations for the dynamics which generate accurate long-horizon predictions (Fig. 5).

**Model-free RL.** A parallel line of work [9, 10, 28] aims to make off-policy model-free algorithms (which form the back-bone for our policy optimization strategy) more stable and efficient in low-data regimes. These approaches introduce regularization techniques which enable the use of higher update-to-data ratios without overfitting to the available data, matching the efficiency of model-free methods such as the MBPO [6] algorithm that we build upon. These algorithmic advances are generally orthogonal to our contribution, and thus in the future we plan to incorporate them into our framework to further accelerate real-world learning.

**Learning Locomotion Strategies in the Real World.** Learning locomotion behaviors from scratch directly in the real-world has primarily been studied in the context of model-free reinforcement learning [45, 46, 32, 47], with a few works using black-box models in the context of model-based reinforcement learning [48, 49]. Compared to these works, our semi-structured modeling approach enable the robot to achieve more dynamic locomotion strategies than these previous approaches, with just a fraction of the real-world samples. Specifically, our approach either achieves a significantly higher walking speed than each of these approaches, or improves on their sample complexity by approximately an order of magnitude (Fig. 4). Several other works investigate fine-tuning locomotion controllers trained in simulation to reduce the burden on real-world data [28, 50] – as we discuss below, we hope to investigate this direction in the near future.

**Direct Transfer From Simulation.** There has also been recent and rapid progress directly transferring locomotion controllers from simulation zero-shot [51, 26, 25, 52, 53], using techniques such as domain adaptation and domain randomization. In this paper we have focused on learning locomotion controllers from scratch, in an effort to demonstrate the ability of our framework to substantially adapt the behavior of the robot with small amounts of real-world data. However, in the future we plan to fine-tune policies that have been trained using extensive simulated experience, improving the performance of these policies in cases where they fail [28] but leveraging a better initialization for the policy for real-world learning.

# 6 Limitations

This paper presents a novel framework for model-based reinforcement learning, which leverages physics-informed, semi-structured dynamics models to enable highly sample-efficient policy learning in the real world. However there are several key limitations. First, our method requires observability of enough proprioceptive states to propagate the Lagrangian dynamics of the robot. Additionally, relying solely on proprioception restricts the model's ability to predict changes to the environment such as the appearance of an obstacle or transitions between different ground surfaces. In the future we plan to extend the current framework to include additional perceptual modalities which can infer more about the state of the environment around the robot.

**Acknowledgments**

The authors would like to thank Trey Smith and Brian Coltin for their helpful insights and feedback. This work was supported by a NASA Space Technology Graduate Research Opportunity under award 80NSSC23K1192, and by the National Science Foundation under Grant No. 2409535.

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

# A   Implementation Details

In this appendix, we provide details of our implementation for the Unitree Go1 Quadruped, including the observation and action spaces, the reward function, the termination condition, and the control architecture.

## A.1   Observation and Action Spaces

The observation space $\Omega \subset \mathbb{R}^{36}$ consists of the elements in Table 1. The $x$-axis of the base is the forward direction, the $y$-axis is the leftward direction, and the $z$-axis is the upward direction. The phase variable $\phi \in [0, 2\pi)$ represents progression along the gait cycle and is defined as $\phi_t = 2\pi t/T_\phi$ mod $(2\pi)$ where $T_\phi = 0.5$ sec is the gait cycle period.

| Observation | Symbol | Dimension |
|---|---|---|
| Quaternion orientation of the base | $\varphi$ | 4 |
| Joint angles | $q^j$ | 12 |
| Base linear velocity (local frame) | $(v^x, v^y, v^z)$ | 3 |
| Base angular velocity (local frame) | $(\omega^x, \omega^y, \omega^z)$ | 3 |
| Joint speeds | $\dot{q}^j$ | 12 |
| Cosine of phase | $\cos\phi$ | 1 |
| Sine of phase | $\sin\phi$ | 1 |

Table 1: Observation space.

The action space $\mathcal{A} \subset \mathbb{R}^9$ outputs the change in nominal height for the gait generator and offsets to nominal foot positions from the gait generator, as defined in Table 2.

| Action | Symbol | Dimension | Min. | Max. |
|---|---|---|---|---|
| $x$-foot position changes | $\Delta p^x$ | 4 | $-0.15\,\text{m}$ | $0.15\,\text{m}$ |
| $y$-foot position changes | $\Delta p^y$ | 4 | $-0.075\,\text{m}$ | $0.075\,\text{m}$ |
| Change in gait generator nominal height | $\Delta h^{\texttt{Gait}}$ | 1 | $-0.1\,\text{m}$ | $0.0\,\text{m}$ |

Table 2: Action space.

## A.2   Reward Function and Termination Condition

**Reward Function.** The reward function is a weighted sum of the terms in Table 3. We set the weights and use exponentials in most of the terms to normalize the reward such that a forward velocity of $1.0\,\text{m}\,\text{s}^{-1}$ with maximal values for all other terms will result in a reward of approximately 1.0 for a single time step. The roll $\varphi^x$, pitch $\varphi^y$, and yaw $\varphi^z$ of the base are obtained from the base quaternion $\varphi$. We define $a \odot b$ as the element-wise multiplication of vectors $a$ and $b$. Actual torques output from the joint-level PD controllers are not available; we estimate the torque applied at the joint with (11). We define the following LinearLimit function which linearly penalizes the torque applied at the $j$-th joint $\tau^j$ when exceeding torque limits; within torque limits, the function is a decaying exponential:

$$\texttt{LinearLimit}(\tau^j, \tau^j_{\texttt{min}}, \tau^j_{\texttt{max}}) = \begin{cases} \tau^j - \tau^j_{\texttt{min}} - 1 & \text{if } \tau < \tau^j_{\texttt{min}} \\ -\exp\left[-\tau^j + \tau^j_{\texttt{min}}\right] & \text{if } \tau^j_{\texttt{min}} \le \tau^j < 0 \\ -\exp\left[\tau^j - \tau^j_{\texttt{max}}\right] & \text{if } 0 \le \tau^j < \tau^j_{\texttt{max}} \\ -\tau^j + \tau^j_{\texttt{max}} - 1 & \text{if } \tau^j \ge \tau^j_{\texttt{max}}. \end{cases} \quad (6)$$

| Reward Term | Expression | Weight |
|---|---|---|
| Maximize forward velocity | $v_{t+1}^x$ | 0.42 |
| Limit base yaw rate | $\exp\left[-(\omega_{t+1}^z)^2/0.2\right]$ | 0.11 |
| Limit base roll | $\exp\left[-(\varphi_{t+1}^x)^2/0.25\right]$ | 0.05 |
| Limit base pitch | $\exp\left[-(\varphi_{t+1}^y)^2/0.25\right]$ | 0.05 |
| Limit base yaw | $\exp\left[-(\varphi_{t+1}^z)^2/0.07\right]$ | 0.11 |
| Limit base side velocity | $\exp\left[-(v_{t+1}^y)^2/0.01\right]$ | 0.11 |
| Limit vertical acceleration | $\exp\left[-(v_{t+1}^z - v_t^z)^2/0.02\right]$ | 0.03 |
| Limit base roll rate | $\exp\left[-(\varphi_{t+1}^x - \varphi_t^x)^2/0.001\right]$ | 0.03 |
| Limit base pitch rate | $\exp\left[-(\varphi_{t+1}^y - \varphi_t^y)^2/0.005\right]$ | 0.03 |
| Limit energy | $\exp\left[-\left\|\dot{q}_{t+1}^j \odot \tau_{t+1}\right\|_1^2/450\right]$ | 0.05 |
| Penalize excessive torques | $\sum_j \texttt{LinearLimit}(\tau_t^j, \tau_{\min}^j, \tau_{\max}^j)/12$ | 0.02 |

Table 3: Reward function terms. The reward at each time step is a weighted sum of these terms.

**Termination Condition.** The termination flag $d_t$ stops the accumulation of reward after the quadruped falls and is defined by:

$$d_t = \begin{cases} 1 & \text{if } |\varphi_t^x| > \pi/4 \text{ or } |\varphi_t^y| > \pi/4 \\ 0 & \text{otherwise.} \end{cases} \tag{7}$$

### A.3 Control Architecture

Here, we give detailed specification of the control architecture introduced in Section 2. Referring to the action space definition Table 2, the policy takes in the current observation and a history of observations and outputs offsets to foot positions and a nominal height for the gait generator: $(\Delta p_t^x, \Delta p_t^y, \Delta h_t^{\texttt{Gait}}) \sim \pi_\theta(\cdot \mid s_t, h_t)$. The gait generator $\texttt{Gait}: [0,1) \times \mathbb{R} \to \mathbb{R}^3$ is open-loop and generates for each leg, walking-in-place foot positions for the quadruped by computing vertical foot position offsets from nominal standing foot positions:

$$\texttt{Gait}(\bar{\phi}_t^l; \Delta h_t^{\texttt{Gait}}) = \begin{cases} p_{\texttt{stand}}^l + \left[0, 0, h^{\texttt{Swing}}\left(1 - \cos\left[2\pi\frac{\bar{\phi}_t^l - r^{\texttt{Gait}}}{1 - r^{\texttt{Gait}}}\right]\right) - \Delta h_t^{\texttt{Gait}}\right] & \text{if } \bar{\phi}^l \geq r^{\texttt{Gait}} \\ p_{\texttt{stand}}^l + [0, 0, \Delta h_t^{\texttt{Gait}}] & \text{otherwise} \end{cases} \tag{8}$$

where $p_{\texttt{stand}}^l \in R^3$ is the nominal standing foot position of the $l$-th leg, expressed in the local base frame, $h^{\texttt{Swing}} = 0.09\,\text{m}$ is the gait peak swing height, and $r^{\texttt{Gait}} = 0.5$ is that fraction of the time feet should remain in contact with the ground. The normalized phase $\bar{\phi}^l \in [0,1)$ specifies the progress of the $l$-th leg along its gait cycle and is calculated with:

$$\bar{\phi}_t^l = \left(\frac{\phi_t}{2\pi} + 0.5 + b^l\right) \mod 1, \tag{9}$$

where $b^l$ is the phase bias for the $l$-th leg; we use a value of 0 for the front-right and rear-left legs, and a value of 0.5 for the front-left and rear-right legs. The desired positions of the $l$-th foot in the local base frame are given by:

$$p_t(\Delta p_t^{x,l}, \Delta p_t^{y,l}, \Delta h_t^{\texttt{Gait}}, \bar{\phi}_t^l) = \left[\Delta p_t^{x,l}, \Delta p_t^{y,l}, 0\right] + \texttt{Gait}(\bar{\phi}_t^l; \Delta h_t^{\texttt{Gait}}), \tag{10}$$

where $\Delta p_t^{x,l}$ and $\Delta p_t^{y,l}$ are the $x$- and $y$-foot positions offsets for the $l$-th foot from the policy. For each foot, the desired foot positions (10) are computed and sent to an inverse kinematics solver to produce desired joint angles $q^{\texttt{des}} \in \mathbb{R}^{12}$. The desired joint angles are sent to the joint level PD controllers, where the desired torque outputs are:

$$\tau_t = K_p(q^{\texttt{des}} - q^j) - K_p\dot{q}^j, \tag{11}$$

and we use proportional gain $K_p = 112\,\text{N}\,\text{m}\,\text{rad}^{-1}$ and derivative gain $K_p = 3.5\,\text{N}\,\text{m}\,\text{s}\,\text{rad}^{-1}$.

# B  Additional Experiments

In this appendix, we present the results of additional experiments which demonstrate our model's ability to make accurate predictions far into the future, generalized to unseen training data, and enable efficient training.

## B.1  External Torque Estimate Validation

Contact forces are discontinuous in nature and can be difficult to learn. To this end, we validate the learned external torque predictions $\bar{\tau}_t^{e,i}$ by comparing them to estimates of the real-world external torques $\tau^e$ over one second of real-world data, presented in Fig. 6. Estimates of the real-world external torque are computed as $\tau^e = M(q_t)\ddot{q}_t + C(q_t, \ddot{q}_t) + G(q_t) - B\tau_t$ where joint accelerations are estimated by finite differencing: $\ddot{q}_t = (\dot{q}_{t+1} - \dot{q}_t)/\Delta t$ and low-level motor torques $\tau_t$ are estimated with the PD control law. The external forces acting on the floating base and the external torques acting on the joints of the front-right leg are shown. Our learned external torque predictions closely align with the estimated actual external torques. Notably, the predictions appear as smoothed versions of the actual torque estimates. Finite differencing of the velocity measurement introduces noise into the actual torque estimates while our multi-step loss learning process inherently smooths the predictions.

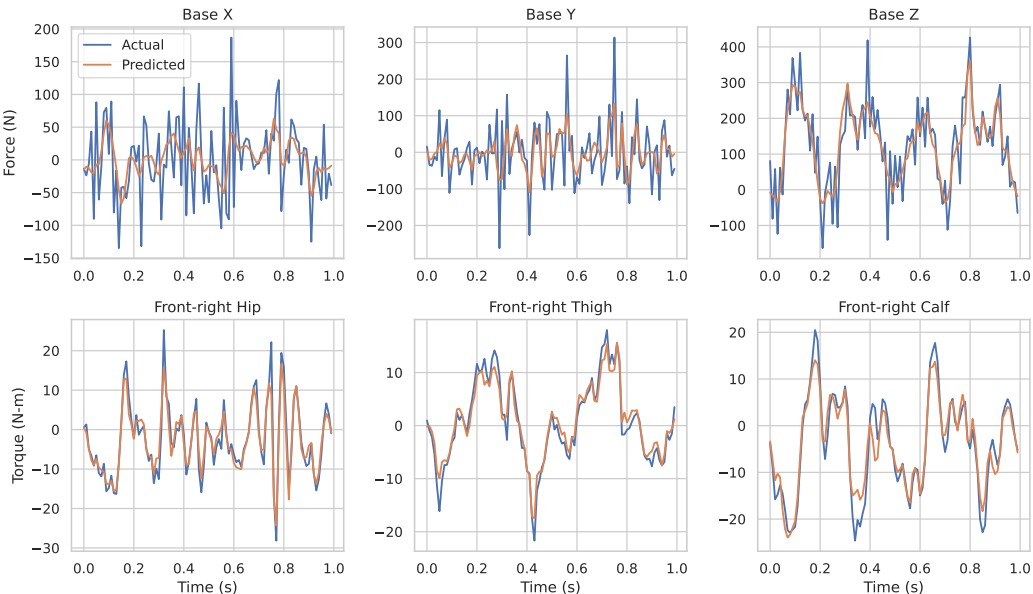

Figure 6: Predicted external forces and actual external force estimates over one second of real-world data for the floating base and the joints of the front-right leg. Our learned external torque predictions closely align with the estimated actual external torques.

## B.2  Model Rollout Accuracy

Here, we examine the accuracy of trajectories generated from learned models and the ability of the model to generalize beyond the available training data. First, we train our semi-structured models and the black-box models from scratch over 3 minutes of saved simulated data using 1- and 4-step losses. Using the trained models, 20-step synthetic rollouts are generated from 400 randomly-sampled starting states within the data. The average prediction error $\|\hat{s}_t - s_t\|/\dim(s_t)$ at each time step $t$ is averaged over the 400 trajectories. This is repeated for 4 random seeds where the best results from the 1- or 4-step losses for each model type are recorded. We then repeat this experiment for the real-world dataset. Finally, the experiment is repeated again, except we generate

a new simulated dataset using a stochastic policy and on altered terrain: friction and ground contact stiffness are lowered by 25%. For this final case, the saved models from the first case are evaluated on this new dataset. The results for these 3 experiments are presented in Fig. 7. For all cases, our semi-structured models produce predictions 20 steps into the future that are significantly more accurate than black-box models. The results from the last experiment demonstrate our model's superior ability to generalize to an unseen environment where noise is added to actions.

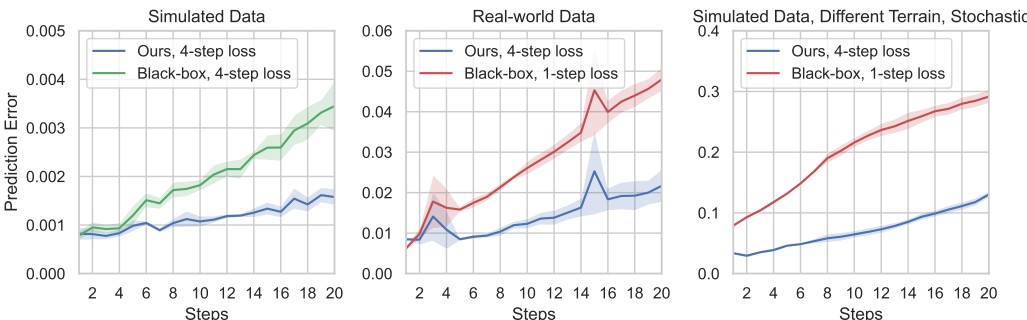

Figure 7: Prediction error for 20-step synthetic rollouts using our semi-structured dynamics models and the black-box models where the best results from the 1- or 4- step losses are presented. Prediction error is averaged over 400 trajectories. Plots show the mean and standard deviation over 4 random seeds.

## B.3 Robustness to Errors in the Lagrangian Dynamics

In this experiment, we demonstrate that SSRL performance is robust against errors in *a priori* knowledge of the robot's inertial properties which are used to construct the Lagrangian dynamics (1). To simulate these modeling errors, we randomly vary each link's mass by ±25% and each joint's damping by ±50% for the Go1 environment used for simulated data collection. The remainder of the setup is per Section 4.2 and runs are repeated across 4 random seeds. The results are presented in Fig. 8; even when there are errors in the *a priori* knowledge of the robot's inertial properties, policy performance is similar to runs with no errors. Our external torque estimators (Section 3.1) learn to predict forces to compensate for these errors, resulting in the similar performance.

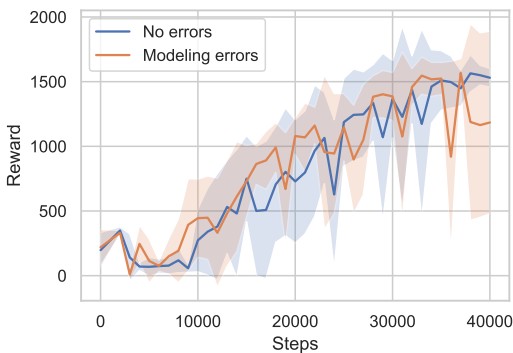

Figure 8: Our approach is robust to errors in *a priori* knowledge of the robot's inertial properties.

## B.4 Modeling Uncertainty

Here, we examine the benefit of using the noise estimator and an ensemble of models. The noise estimator prevents overfitting to noise and using an ensemble of models captures epistemic uncertainty

[27]. We plot the training performance over 4 random seeds for 3 test cases: (i) using the probabilistic ensemble described in Section 3.1, (ii) removing the noise estimators, and (iii) removing the noise estimators and the ensemble. The results are presented in Fig. 9; the highest performance is obtained with our probabilistic ensemble.

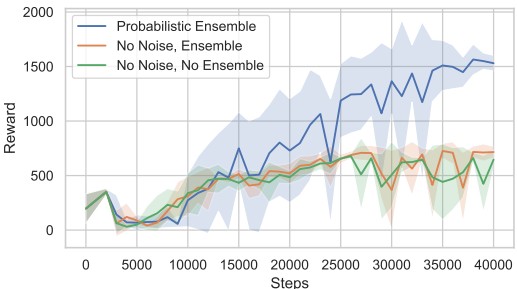

Figure 9: Training performance when removing the noise estimators and removing both the noise estimators and ensemble.

## B.5 Additional Simulated Terrain Experiments

To further demonstrate the versatility of our approach on varying contact surfaces, we perform additional experiments in simulation. Within the simulator, we vary the friction coefficient and contact time constant which determines the stiffness of contact. We perform additional training runs for friction coefficients of 0.3 and 0.8, and contact time constants of 0.06 and 0.12; the friction and time constant used for all other simulated experiments in this paper are 0.6 and 0.02 respectively. As depicted in Fig. 10, we find that our method still yields favorable performance even under varying simulated contact conditions.

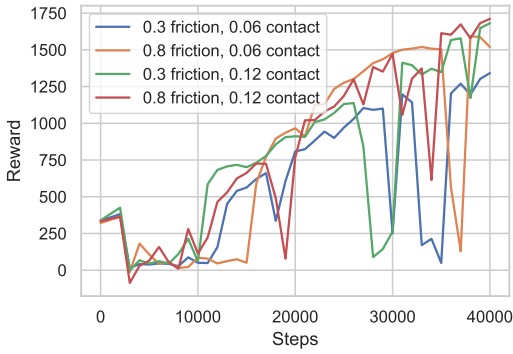

Figure 10: Training performance of our method with varying simulated contact conditions.

## B.6 SAC Performance

State-of-the-art real-world quadrupedal locomotion results where policies are trained from scratch [32] utilized SAC-like agents which directly take actions in the environment. However, highly restricted action spaces were required to obtain stable training behavior with 20 minutes of training data. In contrast, we use action spaces which are more in line with standard quadruped results [25, 29, 30]. This enables faster and more dynamic gaits, but also leads to a challenging optimization problem for SAC. Here, we repeat the experiment of Section 4.2, but we allow the SAC training to run for many more samples and present the results in Fig. 11. Even though SAC now converges, it takes 250 times more interaction with the environment than our method.

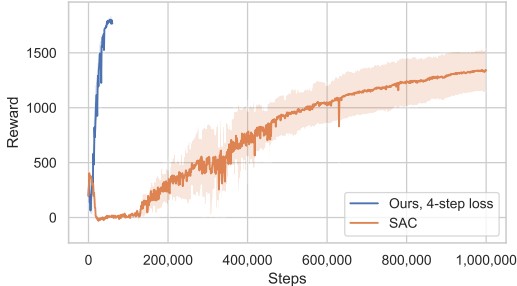

Figure 11: Training curves when running SAC to convergence; SAC requires 250 times more interaction with the environment than our method.

## C   Simulated Benchmark Experiments

To demonstrate the versatility of our approach, we perform additional simulated experiments using standard, contact-rich, benchmark environments [54] commonly used to evaluate reinforcement learning (RL) algorithms.

**Experimental setup.** We use the standard MuJoCo [54] environments Hopper, Walker2d, and Ant, which have been implemented as part of Brax [37]. Similar to the quadruped, each of these environments feature a floating-base robot with articulated limbs which make and break contact with the ground to produce motion. However, unlike the Go1 environment, these environments lack structured controllers. Instead, the outputs from the policy are only scaled linearly before being directly applied as torques on the joints. To test Hypothesis 1 and Hypothesis 2, we compare our semi-structured approach trained with a multi-step loss ($H = 4$) to the black-box approach from Section 4.2 trained with the single-step loss ($H = 1$). In both of these cases, the agent acts deterministically within the environment per Algorithm 2. We also benchmark against SAC [33], allowing the agent to act stochastically in the environment for this algorithm only. The hyperparameters used for training are found in Appendix D and all runs are repeated for 4 random seeds.

**Results.** The results of these experiments are presented in Fig. 12. We observe a significant performance improvement when utilizing our semi-structured models trained with a multi-step loss, compared to the black-box approach trained with a single-step loss, confirming Hypothesis 1 and Hypothesis 2. These results demonstrate that our approach works not only with the Go1 environment, but also with other contact-rich environments with unstructured controllers.

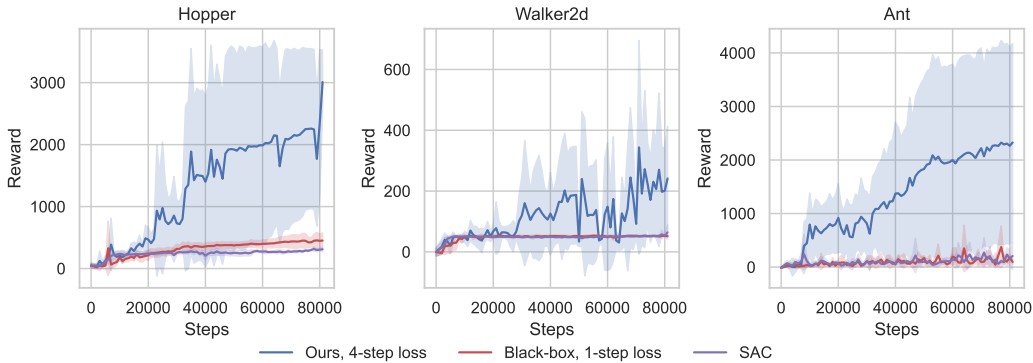

Figure 12: Simulated benchmark results. Better performance is achieved when using our semi-structured dynamics models and a multi-step loss. Plots show the mean and standard deviation for episodic rewards.

# D Experiment Hyperparameters

Table 4 contains the hyperparameters used with our approach; these hyperparameters were also used with the approach that incorporated black-box models. Table 5 contains the SAC hyperparameters used for our approach, the black-box approach, and standard SAC.

| Hyperparameter | Go1 (real world) | Go1 (simulated) | Benchmarks |
|---|---|---|---|
| Epochs, $N_{\text{epochs}}$ | 18 | 40 | 80 |
| Environment steps per epoch, $N_E$ | 1000 | | |
| Hallucination updates per epoch, $K$ | $10 \to 1,000$ over epochs $0 \to 4$ | | |
| Model rollouts per hallucination update, $M$ | 400 | | |
| Synthetic rollout length, $k$ | $1 \to 20$ over epochs $0 \to 10$ | | $1 \to 45$ over epochs $0 \to 15$ |
| Real to synthetic data ratio, $r_{\mathcal{D}}$ | 0.06 | | |
| Gradient updates per hallucination update, $G$ | 40 | 60 | 20 |
| State history length, $h$ | 5 | | 1 |
| Multi-step loss horizon, $H$ | 4 | 1 or 4 | |
| Model learning rate | $1 \times 10^{-3}$ | | |
| Model training batch size | 200 | | |

Table 4: Hyperparameters for our approach and the baseline approach with black-box models. $x \to y$ over epochs $a \to b$ denotes a clipped linear function, i.e. at epoch $i$, $f(i) = \texttt{clip}(x + \frac{i-a}{b-a}(y - x), x, y)$.

| Hyperparameter | Go1 (real world) | Go1 (simulated) | Benchmarks |
|---|---|---|---|
| Learning rate | $2 \times 10^{-3}$ | | $3 \times 10^{-3}$ |
| Discount factor, $\gamma$ | 0.99 | | |
| Batch size | 256 | | |
| Target smoothing coefficient, $\tau$ | $1 \times 10^{-3}$ | | $5 \times 10^{-3}$ |
| Actor network (MLP) width $\times$ depth | $512 \times 2$ | | $256 \times 2$ |
| Critic network (MLP) width $\times$ depth | $512 \times 2$ | | $256 \times 2$ |

Table 5: SAC hyperparameters used for our approach, the black-box approach, and standard SAC.

