# OpenReview forum: "Learning to Walk from Three Minutes of Real-World Data with Semi-structured Dynamics Models"
_robot-learning.org/CoRL/2024/Conference — CoRL 2024_

### Official Review · Reviewer_reP2 · 2024-07-16
**Impressive hardware results and a strong direction for learning in the real world.**

**Originality:** 2
**Technical Quality:** 4
**Clarity Of Presentation:** 4
**Potential Impact:** 3
**Recommendation:** 4
**Confidence:** 5

**Review:**

Overall I think the work is good and presents a clear idea forward that can be built upon. Though there have been other learning in the real world and model-based RL for quadruped locomotion before (which the authors have appropriately referenced), I have not seen it structured through Lagrangian dynamics as it is here before. While I am not fully convinced of the full usefulness of this method yet, i.e. I am not convinced enough that I would use this method myself to train the locomotion policies I want (see below for more details), this is definitely a step in the right and interesting direction, and I am definitely interested in extension ideas that stated in the limitations (adding in external perception) and appendix (fine tuning simulation trained policies in the real world).

## Strengths:

This paper tackles an important topic highly relevant to the field: how to best incorporate real world data into the learning process and how to learn from a small amount of real world interactions. Solving this would help many a sim2real problems that I’m sure a lot of researchers face currently.

The work is overall well written and clear. The method is presented well and is easily understandable; after reading I feel like I understand the main idea enough to know how I would implement/use this myself. The theoretical side of things seem to be sound to my knowledge as well.

Ablation studies are included to justify certain design choices as well when it comes to the resulting policy performance.

The biggest strength by far are the impressive hardware results. Using only 3 minutes of hardware is highly impressive and is a real world sample efficiency record for quadruped locomotion to the best of my knowledge. This along with the relative soundness of the method makes this an accept for me regardless of the limitations described below.

## Limitations:

The simpleness of the quadruped locomotion test case used here weakens how convinced I am of the actual usefulness of the presented method. I understand that this is just a test case to show your proposed method but it still stands that nowadays quadruped locomotion is mostly “solved” and is sort of a given capability. Nowadays our simulation models are good and fast enough that I’m sure in the wall clock time spent to learn the dynamics model and policy in your method you can just learn a policy in simulation and zero-shot transfer to the real hardware. Or even if sim2real is an issue, adaptation methods like RMA seem to be able to fix things with small amounts of real world data as well.
I am much more interested in seeing how your method would scale up to harder and more complex tasks/platforms like non statically stable bipeds, where the dynamics might be harder to model, learning on hardware is more complicated, and highly structured policy action spaces like footstep gait controllers are harder to engineer (which I think is highly important for both the sample complexity and feasibility of taking exploratory actions on hardware done in this work).

The main novelty here is relatively limited, and seems to be just the structuring of the model learning as Lagrangian dynamics. Everything else, as the authors already reference too, has been done before. Non black box model based RL has also sorta already been done before, see SimGan (Jiang et. al, ICRA 2021), where they learn modification to simulation parameters rather than a whole black box (though they admittedly use the learned model for policy refinement rather than learning from scratch).

A major part of this work is learning the semi-structured dynamics model, but there is not evaluation of the model’s performance itself, only how it affects policy performance. Some evaluation of how accurate the learned model is (or average of the ensemble of learned models) in your simulated results would be very good to add I think. Along with some model accuracy comparisons with the black-box approach and the single step loss, especially since you state that the goal is to do auto-regressive predictions “far into the future” and that your method can “produce rollouts that generalize far beyond the available training data”. There are currently no evaluations that attempt to show this, only evaluations of policy performance.

## Formatting comments:

Line 65/66: “but is required” is repeated in the sentence.

Line 159: “... a predictive model for the howe the environment…”

Line 245: “environment” misspelled as “nvironment”

Figure 5 caption: “semi-structed dynamics models”, “structured” is misspelled

**Quality Of The Limitations Section:**

3

**Questions For Rebuttal:**

What’s the rationale for structuring things as Lagrangian dynamics and learning the external force component as opposed to just learning some black box modification to the simulator output, whether it be to simulator parameters or just a straight up addition to the simulator state? To me it seems like the simulator’s estimate of the next state would be better than Lagrangian dynamics (simulator’s output is probably closer to the real world next state than the Lagrangian’s estimate without the external force component, so you have to learn a less difficult problem) and would probably actually be my first thought of what to do rather than this Lagrangian setup.

As the authors point out the method requires being able to differentiate through the mass matrix and Coriolis terms in order to optimize the probabilistic dynamics model. This work gets around this by using Brax, a differentiable physics simulator, but this requirement may be a sticking point for some setups which do not have this capability. How accurate of a gradient is required of this? Would something like finite differencing through a non-differentiable simulator work? Do you have any intuition on how this affects the accuracy of the learned dynamics model, and how does the accuracy of the dynamics model affect the learned policy?

For the simulation experiments, why does SAC not learn? I would expect it to still learn, just more slowly, but it seems like the learning actually collapses? For the other black box and 1-step loss comparisons I can understand it failing since a badly learned model can cause the policy learning to go bad, but SAC is taking actions directly in the environment. For example, as you reference (ref. 45) other works can learn to walk in the real world with SAC with 30000 samples. Why does your SAC baseline seem to underperform?

**Robotics Focus:**

4

**Summary Of Paper:**

The paper proposes a model based reinforcement learning technique that structures the model learning problem as Lagrangian dynamics and learns just the external forces term as opposed to the whole state transition function. By exploiting the prior knowledge of the model through this structure, along with other techniques like learning an ensemble of models and using a multi-step prediction loss, allows the authors to learn locomotion skills on a quadruped from just 3 minutes of real world data. Hardware results are presented with the robot walking both hard and soft ground, as well as simulation ablation studies showing the usefulness of the semi-structured dynamics model and the multi-step loss.

**Summary Of Recommendation:**

Relatively incremental work when it comes to novelty, but I think it is working towards a good direction to explore in for doing learning from limited real world data. The results can be a benchmark for real world sample efficiency, showing successful locomotion learning from the smallest amount of real world data I have seen yet.

---

### Official Review · Reviewer_bENK · 2024-07-20
**A model-based RL framework that integrates a first-principle model with a learned residual dynamics model to enhance quadruped robot locomotion.**

**Originality:** 2
**Technical Quality:** 2
**Clarity Of Presentation:** 2
**Potential Impact:** 2
**Recommendation:** 2
**Confidence:** 4

**Review:**

1. The paper is very well written, with clear motivation and structure. It is easy to follow and understand.

2. My main concern is that the proposed framework lacks novelty on the algorithmic side and fails to demonstrate impressive real-world performance. On the model learning side, it is very common to combine a first-principle model with residual model learning. Conceptually, both the model learning and the policy learning are not new. On the demonstration side, I expect the proposed framework to be demonstrated on more complex terrains. The real-world experiments only show results on simple flat ground or memory foam. What about more complex terrains or slippery ground?

3. The title “Learning to Walk from Three Minutes of Data with Semi-structured Dynamics Models” can be misleading. The three minutes of data is used to learn the residual model, not the walking policy. Policy learning requires both simulation data and real-world data. Therefore, this framework requires significantly more data for learning to walk.

4. Generalization is a key advantage of leveraging a prior model. I suggest that the authors focus on demonstrating how the proposed framework could be generalized to different environments. For example, is it possible to train on certain types of terrain and then generalize the policy/model to unseen terrains? To my understanding, both the policy and the model are overfitting the training environment.

**Quality Of The Limitations Section:**

2

**Questions For Rebuttal:**

1. What is the key novelty of the proposed framework?

2. Can the proposed framework be applied to more difficult locomotion tasks?

3. How much data is actually required to train the walking policy? (both simulation and real-world data)

4. Is it possible to train on certain types of terrain and then generalize the policy/model to unseen terrains?

**Robotics Focus:**

4

**Summary Of Paper:**

This paper presents a model-based reinforcement learning framework that integrates a first-principle model with a learned residual dynamics model to enhance quadruped robot locomotion. For floating-base systems like quadruped robots, controlling the un-actuated base coordinates, such as the robot's body, requires external forces. Instead of relying on traditional methods like the soft contact model or kinematic constraints, this paper introduces an approach to learn the external forces based on the observed states of the robots. The model learning process utilizes a neural network to predict the external torques applied to the robot by its environment. The neural network is trained using approximate maximum likelihood estimation. For policy training, the authors employ the Soft Actor-Critic algorithm, utilizing both real-world and simulated data to optimize the policy. The paper demonstrates the proposed approach through real-world results and simulated experiments focused on the task of tracking a forward velocity. Real-world results indicate that the proposed framework can be used to train a neural network control policy for simple quadruped locomotion. The simulation experiments show that the hybrid robot model has better simulation results than a black-box model.

**Summary Of Recommendation:**

The contribution of this work is minor. Weak reject.

---

### Official Review · Reviewer_V9T9 · 2024-07-21
**Cool paper making a point for using mechanistic structural knowledge to improve sample efficiency**

**Originality:** 4
**Technical Quality:** 4
**Clarity Of Presentation:** 4
**Potential Impact:** 4
**Recommendation:** 3
**Confidence:** 4

**Review:**

Currently, with the many recent papers coming out on quadrupedal locomotion, one could think that this problem is solved. However, the data-efficiency of these algorithms is still quite bad which this paper argues can be solved using mechanistic models such as the Lagrange equations. The algorithm proposed by the authors is rather simple which I perceive as strength.

**Strengths:**
- Convincing experimental results.
- Relatively simple model architecture.
- Hardware experiments on quadruped robot with two different ground materials.

**Comments:**
- **C1 - Verbose writing:** The paper reads as if the authors just started writing section after section without thinking in advance how to most effectively communicate its ideas to the reader. After all, the authors just propose to model dynamics by plugging an ensemble model into the joint torques of the Euler-Lagrange equations and adding a second head to the assemble model that adds noise to the models output. This model is than used in MBPO to hallucinate rollouts from the real-world data onto which a modified SAC is being trained. Why not first give a systematic high-level overview and then one by one explain the algorithms components?

- **C2 - Errors in Lagrange equations:** In line 56, the authors write *"$J^T F_e$ is the only unknown term in the dynamics"*. If $\tau$ just corresponds to the motor torques (typically estimated by multiplying motor currents with the motor constant), then the **dissipative torques in the robot's joints remain unmodeled**. Dissipative torques can be quite a nuisance for forward model predictions which is why "Learning agile and dynamic motor skills for legged robots" by Hwangbo et al deploys joint torque sensors in a  pre-identification step to solely identify joint torques. As the authors learn "external forces" in generalized coordinate space, the **model also learns dissipative torques** along side the end-effector torques.

- **C3 - Please share your code and data**

**Minor comments:**
 - Line 54: The force $C$ contains not only Coriolis terms, **but also centrifugal forces**. Featherstone refers to "C" in "Rigid body dynamics algorithms" as "bias term". I like this terminology, as if no external forces act on the system, then $C$ is the force one needs to apply to cause zero acceleration.
 - I would avoid refrain from using *"sophisticated"* words such as "uncertainty-aware" or "proprioceptive" if it is not needed. Just as an example, the authors use the word **"proprioceptive"** over seven times in the paper, why?? You have established in the introduction that the states are reprioceptive, why always add a word that does not help in communicating your actual ideas?
- Equation 4 is actually a table, please adjust the formatting. I am doubtful that this table serves the reader as adding the names of these model components to Figure 2 would already do the trick while providing also information on the model's graph.
- In Figure 2, it looks like the output of the ensemble model and the Lagrange equations are being added, but if I understood correctly you plug the output of the ensemble model **into the Lagrange equations**.
 - Line 77, typo "that"

**Quality Of The Limitations Section:**

2

**Questions For Rebuttal:**

- **Question 1:** When determining policy parameters θ with SAC, how do you choose how many transitions to pick from Denv and Dmodel?
- **Question 2:** How do you plan to extend this method to non-flat surface geometries and surfaces covered my movable objects in particular foliage?
- **Question 3:** Why do you need an encoder prior to the ensemble model? Can you add an ablation study that investigates how removing the encoder/noise network/... affects training?
- **Question 4 - Learning contact forces:** Due to the discontinuous nature of contact forces, learning these forces is extremely hard. So, I am surprised that the authors got anything tangible out of the ensemble. Clearly, the ensemble model is learning the contact force over one time step which causes some smoothing. Also the encoder might discover directions in generalized coordinate space in which your dynamics remain invariant to discontinuities (null space of $J^T$, see e.g. "Contact invariant model learning for legged robot locomotion" by Grandia et al). **Can you add another plot that shows the collected real-world data minus the Lagrange model without external torques, and then compare to this data to the predicted external joint torque that has been predicted by the ensemble?** I would be interested in seeing the absolute error of the ensemble errors on the whole data set and also seeing rollouts of the signal over time.

**Robotics Focus:**

4

**Summary Of Paper:**

The authors train a policy that has only excess to a series of past observations of joint encoders and IMU measurements. The policy outputs the parameters of repetitive motion generator and offsets to nominal foot position.  This motion generator then outputs foot positions which via an IK solver are mapped to reference joint positions that are tracked by a PID. During training the policy is used to collect data which is then used to train an ensemble of probabilistic dynamics models that is supposed to predict the external forces (in particular contact) acting on the robot. These external force predictions (being in generalized coordinate space and being samples of the model) are then inserted into the robot's Euler-Lagrange equations to hallucinate possible motion. On this hallucinated data, the policy is trained via SAC.

**Summary Of Recommendation:**

Solving a difficult problem with few means is a significant feat. If the authors work on the structure of this paper, its writing style, add few additional experiments, and extend the critical discussion, than this work is **top notch**.

---

### Author Rebuttal · Authors · 2024-08-10

We would like to thank the reviewers for their careful reviews of the submission. We have incorporated their feedback into a new draft attached to this rebuttal.

---

### Decision · Program_Chairs · 2024-09-04

**Decision:**

Accept

**Comment:**

**Post-rebuttal metareview**:

This paper proposes an MBRL approach for locomotion, where a residual dynamics model is learned on top of Lagrangian dynamics, and a SAC policy is trained using the learned dynamics. This method is quite novel in the field of locomotion, and justified by hardware experiments. Applying the proposed techniques in more challenging tasks (e.g., complex terrain) and showing its advantage over model-free approaches will further strengthen the paper.

----------------
**Pre-rebuttal metareview**:

Strengths:
1. The idea of using physics prior knowledge for efficient MBRL in locomotion is interesting and novel.
2. Convincing experimental results to prove the concept.

Weakness:
1. The presentation needs improvements.
2. Some claims are not accurate such as “the external contact force is the only uncertainty”.
3. Although relatively new in locomotion, the key idea is not new in robotics.
4. Need more challenging experiment scenarios such as rough terrains.
5. Other issues mentioned by reviewers, such as generalization, definition of “3-min data”, etc.